# Characteristics of P-Type and N-Type Photoelectrochemical Biosensors: A Case Study for Esophageal Cancer Detection

**DOI:** 10.3390/nano11051065

**Published:** 2021-04-21

**Authors:** Joseph-Hang Leung, Hong-Thai Nguyen, Shih-Wei Feng, Sofya B. Artemkina, Vladimir E. Fedorov, Shang-Chin Hsieh, Hsiang-Chen Wang

**Affiliations:** 1Department of Radiology, Ditmanson Medical Foundation Chia-yi Christian Hospital, Chia Yi 60002, Taiwan; 01289@cych.org.tw; 2Department of Mechanical Engineering, Advanced Institute of Manufacturing with High Tech Innovations (AIM-HI), Center for Innovative Research on Aging Society (CIRAS), National Chung Cheng University, 168, University Rd., Min Hsiung, Chia Yi 62102, Taiwan; nguyenhongthai194@gmail.com; 3Department of Applied Physics, National University of Kaohsiung, 700 Kaohsiung University Rd., Nanzih District, Kaohsiung 81148, Taiwan; swfeng@nuk.edu.tw; 4Nikolaev Institute of Inorganic Chemistry, Siberian Branch of Russian Academy of Sciences, 630090 Novosibirsk, Russia; artem@niic.nsc.ru (S.B.A.); fed@niic.nsc.ru (V.E.F.); 5Department of Natural Sciences, Novosibirsk State University, 1, Pirogova Str., 630090 Novosibirsk, Russia; 6Department of Plastic Surgery, Kaohsiung Armed Forces General Hospital, 2, Zhongzheng 1st. Rd., Lingya District, Kaohsiung 80284, Taiwan

**Keywords:** esophageal squamous cell carcinoma, transient photocurrent, photoelectrochemical biosensor, glutathione

## Abstract

P-type and N-type photoelectrochemical (PEC) biosensors were established in the laboratory to discuss the correlation between characteristic substances and photoactive material properties through the photogenerated charge carrier transport mechanism. Four types of human esophageal cancer cells (ECCs) were analyzed without requiring additional bias voltage. Photoelectrical characteristics were examined by scanning electron microscopy (SEM), X-ray diffraction (XRD), UV–vis reflectance spectroscopy, and photocurrent response analyses. Results showed that smaller photocurrent was measured in cases with advanced cancer stages. Glutathione (L-glutathione reduced, GSH) and Glutathione disulfide (GSSG) in cancer cells carry out redox reactions during carrier separation, which changes the photocurrent. The sensor can identify ECC stages with a certain level of photoelectrochemical response. The detection error can be optimized by adjusting the number of cells, and the detection time of about 5 min allowed repeated measurement.

## 1. Introduction

Reduced glutathione (GSH) is an endogenous tripeptide synthesized from three amines, namely, cysteine, glutamine, and glycine. GSH, which is derived from physiological fluids of mammals, is an ingredient used as a coenzyme in oxidation–reduction reactions in intracellular environments or tissues of organisms. Detoxification is a phenomenon when a reducing agent, namely, sulfhydryl (-SH) formed from GSH, effortlessly reacts with cytotoxins [1,2]. Detoxification is performed to destroy free radicals, reduce mediators of redox reactions, activate enzymes, protect cells from oxidative stress, promote protein and DNA synthesis, and enhance cell metabolism [3]. GSH plays an important role in living organisms because it provides considerable biological information. An abnormality in GSH concentration in biological fluids or tissues is often directly associated with several medical diseases, including diabetes [4,5,6,7], cardiovascular diseases [8,9,10,11], and cancers [12,13,14,15]. GSH is found throughout cellular compartments in millimolar concentrations (1–10 mM). Relative GSH and GSSG concentrations vary slightly across cancer cell lines [16]. Reference [17] shows concentration of GSSG as a function of the concentration of GSH in the same cell line. A ratio of GSH/GSSG across 928 cell lines was equal to one [17]. Therefore, the determination of GSH concentration is necessary. Methods of GSH analysis include colorimetry [18], spectrofluorimetry [14,19], and magnetic resonance spectrometry [20,21,22]. However, these methods exhibit many limitations, such as low sensitivity, long detection time, and time-consuming operation.

Emerging photoelectrochemical (PEC) analysis with a bio-based platform has been widely applied in research because this method offers low cost, high sensitivity, fast detection, and low background current [23,24]. GSH/GSSG is the most abundant redox-active molecule in a cell and plays a crucial role against oxidative stress. Furthermore, the oxidation–reduction degree of GSH/GSSG is an important indicator for many diseases, such as cancers, Alzheimer’s, and neurodegenerative disease. Hence, the monitoring of GSH/GSSH has gained considerable attention, and recent papers have published several innovative PEC sensors for GSH/GSSG detection [25,26,27,28,29]. The analysis of PEC biosensing is based on the principle that when light is used as an excitation agent, it induces changes in the current or voltage in the electrodes as the output signal [23,30]. The expression of PEC is highly dependent on the properties of material constituting the biosensors. Materials used as transducers in PEC biosensors can be classified into inorganic semiconductors [31,32,33], organic semiconductors [19,34,35], and hybrid semiconductors [30,36]. The performance of PEC sensors may vary depending on the properties of synthesized materials; as such, researchers have focused on designing novel materials. Among combinations of materials, heterojunction exhibits outstanding electrical and optical properties. The heterojunction PEC biosensors, as p-type and n-type films, can easily induce photoelectrochemical reactions and has outstanding reproducibility [37,38,39,40,41,42]. In 2018, Zang et al. [43] developed a PEC sensor based on the heterojunction effect of CdS/WS_2_; the sensor achieved approximately a 310% increase in photocurrent response due to CdS quantum dots that modified the heterostructure. Liu et al. [44] successfully synthesized a core-shell unique structure of a graphene PEC sensor for the electrochemical detection of GSH. The prepared sensor exhibited high sensitivity, fast response, short detection time, self-power capability, and satisfactory accuracy for GSH serum. In 2019, Yu et al. [45] modified an electrode with heterostructures of CuO and Cu_2_O; these structures were made of a composite material with photoelectric properties. The structures exhibited excellent biosensing performance in terms of high sensitivity, selectivity, and stability. In 2020, Li et al. [46] developed a PEC sensor by using nanoflakes of MXene-NiFe; the sensor showed excellent charge transport properties and achieved satisfactory results for detecting GSH.

The mortality rate due to esophageal cancer has gradually increased in recent years due to variations in eating habits and living environments [47]. Cancer can be prevented at an early stage; however, most cancers are diagnosed after they reach severe stages or even metastasize, during which treatment is difficult. The rapid introduction of nanotechnology has affected the development of biosensors; the exploration of novel designs of biosensors for diagnosis can improve the prognosis of cancer [48,49]. In the present study, photoelectrochemical analysis was used to discuss the relationship of electron–hole transmission with P-type and N-type structures. A biosensor platform was developed and used to analyze and detect the level of GSH in esophageal squamous cells.

## 2. Materials and Methods

### 2.1. Preparation of P-Type Biosensor

The ITO glass substrate (ITO-GL, Rui Long Optoelectronics, Miaoli, Taiwan) used in this experiment has a resistance value of 7 Ω and a coating thickness of 200 nm. Cuprous oxide (Cu_2_O) film was grown on the ITO glass substrate by electrochemical deposition (ECD). Copper sulfate pentahydrate (CuSO_4_·5H_2_O) (102790, Merck Millipore, Hsinchu, Taiwan) with a purity of 98–102% and 97% pure sodium hydroxide (NaOH, 106462, Merck Millipore, Taiwan) was mixed with deionized water to form an aqueous solution. The two aqueous solutions were mixed to form a PH12 electrolyte, which requires the addition of 85% lactic acid to prevent the electrolyte. We used DC power supply (LPS505N-MO, Motech, Tainan, Taiwan) for the two-pole electrochemical deposition system. 

Molybdenum disulfide (MoS_2_) was grown on a sapphire substrate by chemical vapor deposition (CVD). Then, 99.95% pure molybdenum trioxide powder (0.0005 g, 203815, Sigma-Aldrich, Hsinchu, Taiwan) and 99.999% pure sulfur powder (0.8 g, 43766, Alfa Aesar, Hsinchu, Taiwan) were weighed and placed in the center of the wafer boat. The cleaned sapphire substrate was placed upside down on a wafer boat of molybdenum trioxide powder. The silicon substrate was used to increase the sapphire substrate and facilitate the reaction of the two precursors to obtain high-quality single-layer molybdenum disulfide. Finally, the MoS_2_ sample was coated with Poly methyl methacrylate (PMMA) (AR-P679.04, Allresist, Hsinchu, Taiwan) and heated at 100 °C for 30 min. MoS_2_ was separated from the sapphire substrate by using sodium hydroxide solution with a concentration of 2 M (11963233, Fisher Scientific, Hsinchu, Taiwan) [50]. The copper sample was adhered by heating. After removing PMMA with acetone, P-type biosensor was obtained. Figure 1a shows the flow chart of preparation of P-type biosensor components.

### 2.2. Preparation of N-Type Biosensor

Cuprous oxide (Cu_2_O) films were deposited by electrochemical deposition on indium tin oxide (ITO)-coated substrates. A zinc oxide (ZnO) seed layer was acquired on cuprous oxide by spin coating. A colloidal solution of 99% pure zinc acetate [Zn(CH_3_COO)_2_] (AC370080250, Fisher Scientific, Taiwan) and ethanol with a molar concentration of 0.02 M was prepared, placed on an ultrasonic oscillator for 5 min, and heated in air atmospheric pressure at 350 °C for 30 min. ZnO nanorods were prepared by hydrothermal growth method. Zinc nitrate [Zn(NO_3_)_2_] (99%, 11316158, Fisher Scientific, Taiwan) and hexamethylenetetramine (C_6_H_12_N_4_, 99%) (11387918, Fisher Scientific, Taiwan) with deionized (DI) water were dissolved into two solutions with equal volumes and with molar concentrations of 0.314 and 0.7 M, respectively. The substrate coated with the ZnO seed layer was rinsed with the two solutions placed on a rotating holder at 90 °C for about 2 h and subjected to the hydrothermal growth method. Figure 1b shows the flow chart of the preparation of N-type biosensor components.

### 2.3. Cell Culture

Two types of esophageal squamous carcinoma cells, including, OE21, and healthy cells, were used and incubated in RPMI and DMEM. The Roswell Park Memorial Institute (RPMI, Buffalo, NY, USA) culture medium was used for normal esophageal OE21 cells from a white man, as well as for OE21-1 cancer cells (exhibiting the higher invasiveness in white people). Subpopulations from the CE81T ECC line were selected using the membrane invasion culture system (MICS) or BD BioCoat™ Matrigel™ Invasion Chamber (Corning, NY, USA). Furthermore, 10% fetal bovine serum (FBS, Gibco, Grand Island, NE, USA) and 1% penicillin (Gibco, Grand Island, NE, USA) were added to the culture media. The sampled cells were cultured on Petri dishes (Falcon, Franklin Lakes, NJ, USA) and placed in an incubator at 37 °C containing 50% carbon dioxide. Culture media were replenished, or cells in Petri dishes were sorted, every two days as appropriate. In addition, phosphate buffered saline (PBS, pH 7.4, Biochrom, New York, NY, USA) was used for rinsing while sorting the cells on Petri dishes. Then, 0.25% trypsin and 0.02% EDTA (Sigma, New York, NY, USA) were added to the cell mixture after 5 min to obtain a cell suspension.

### 2.4. Optical and Material Analysis Instruments

Figure 2 shows the images of P-type and N-type biosensors taken using an optical microscope (MM40, Nikon, Lin Trading Co., Taipei, Taiwan) and a cold field-emission scanning electron microscope (FESEM, S4800-I, Hitachi, Tokyo, Japan). The magnification of the optical microscope is about 1500 times, the magnification of the scanning electron microscope can reach more than 10,000 times, and the depth of field is large. It is mainly used to observe the microstructure of the sample surface and section. X-ray diffraction (XRD, Bruker Smart APEX, Bruker, Hsinchu, Taiwan) analysis is a non-destructive analysis technique used to detect the characteristics of crystal materials. It provides structure and phase crystal orientation and parameter analysis (such as average particle size, crystallinity, tension, and crystal defects). The X-ray diffraction peak is the constructive monochromatic light diffracted by the crystal lattice plane of the sample at a specific angle. Produced by interference, the peak intensity is determined by the distribution of atoms in the lattice.

## 3. Results and Discussion

### 3.1. Material Structure Characteristics

#### 3.1.1. Optical Microscope (OM) and Scanning Electron Microscope (SEM) Images

Figure 2a presents the OM image of MoS_2_ grown on the sapphire substrate. The sapphire substrate has a single layer of MoS_2_ with a high concentration. The upper right illustration is a high-magnification view of MoS_2_. The length of the triangle can reach 90 µm. Figure 2b demonstrates the SEM image of MoS_2_ transferred onto the Cu_2_O film. MoS_2_ was successfully transferred on Cu_2_O, and the size was about 45 µm. The upper right illustration shows the high-magnification view of MoS_2_. Figure 2c is the cross-sectional view of the ZnO/Cu_2_O heterostructure. The thickness of the Cu_2_O layer was about 5 µm, and that of the ZnO layer was about 2 µm. Figure 2d shows the SEM top view of ZnO nanorods.

#### 3.1.2. X-ray Diffraction (XRD) Analysis

Figure 3 shows the X-ray diffraction patterns of P-type and N-type biosensors. Cu_2_O is a stable phase in dry air (below air humidity 30%) and is usually stored in a vacuum ball because it will slowly oxidize into copper oxide (CuO) in humid air [51]. However, Cu_2_O is exposed to humid air during the dropping of the cell solution or cleaning of the wafer. The oxidization phases from Cu_2_O into CuO were identified by comparison of the measured diffraction peaks with the cards given by the Joint Committee of Powder Diffraction Standards (JCPDS). Figure 3a shows that the lattice diffraction peaks of Cu_2_O appeared at 2θ = 29.4°, 36.3°, 42.2°, 61.2°, and 73.4°. The oxidation phases of CuO were detected at 2θ = 35.1°, 38°, 50.5°, and 77.2°, corresponding to (110), (111), (200), (220), and (311) lattice directions from the JCPDS card. The diffraction peaks of CuO occurred at (111), (111), (202), and (222) lattice directions from the JCPDS card.

In Figure 3b, the diffraction peaks of the ITO glass substrate were used to remove the background value, while the diffraction peaks of ZnO nanorods were identified to the ZnO crystalline phase and the ZnO/Cu_2_O heterostructure. The angle range was 10°–80°. The lattice diffraction peaks of Cu_2_O appeared at 2θ = 29.6°, 36.5°, 42°, 61.5°, and 73.7°. From the JCPDS card, at (110), (111), (200), (220), and (311) lattice directions, when at 2θ = 34.4°, 36.3°, 47°, 62°, and 67°, the diffraction peaks of ZnO that appeared were compared with those from the JCPDS card at (002), (101), (103), (112), and (201) lattice directions. The XRD patterns showed no diffraction peaks of impurities.

#### 3.1.3. Absorption Analysis

Figure 4 shows the UV–VIS reflectance spectra of P-type and N-type biosensors. Figure 4a shows transparent light absorption of 300–500 nm when measuring cuprous oxide samples through ultraviolet–visible light absorption spectroscopy (Evolution 220, Thermo Fisher Scientific, Waltham, MA, USA). From the MoS_2_/Cu_2_O curve, the light absorbance rate at the wavelength of about 500 nm increased due to the material properties of the molybdenum disulfide after being transferred to the Cu_2_O surface. Previous studies showed the light absorption of a single layer of MoS_2_ for the wavelength range below 680 nm and direct energy gap at 1.9 eV [52]. In this heterogeneous structure of synthetic ZnO and Cu_2_O, light absorption mainly occurred on the Cu_2_O layer. Zinc oxide has light absorption in the UV range and weak light absorption in the visible light range. For the Cu_2_O material layer, the light absorption capability was greatly extended to the visible light region [53]. Figure 4b shows that in the visible light range of 380–780 nm, the Cu_2_O layer increased the light absorption range. The intrinsic material properties of Cu_2_O include its considerable surface roughness of about 1um. In order to successfully transfer the 1nm MoS_2_ film on the Cu_2_O surface, we used mechanical grinding to polish the surface of Cu_2_O. A flat surface will not be conducive to catching light. This results in the lower absorbance of MoS_2_/Cu_2_O than ZnO/Cu_2_O. This structure is advantageous for effectively promoting carrier separation and improving the sensitivity of the photoelectrochemical sensor.

### 3.2. Photoelectrochemical Response Analysis

The three-stage changes in the photoelectrochemical biosensor are as follows. In the first stage, the element is excited by light after being illuminated. A large number of electron–hole pairs are generated, and the current measured by the photoresponse system immediately increases. In the second stage, the photoelectric charge generated by the excitation elements is recombined due to the effect of time. The slope of current is likely to be slow; however, in general, the slope is positive because the generation rate of electron–hole pairs is greater than the recombination rate. In the third stage, after the light is turned off, the component is no longer excited by light, and no electron–hole pair is reproduced. The electron–hole pair is affected by the survival time and disappears. Therefore, the current measured by the light response system begins to decrease. Given that the photogenerated charge is trapped in the deeper defect state in the Cu_2_O structure, it is forced to cross more trapping energy levels before carrier recombination can be performed, resulting in the slow decay of the turn-off photocurrent value [54].

Figure 5a,b show the recovery time of decay current after the component is periodically illuminated. In Figure 5a, when the light is turned on for 20 s, it must be turned off for 50 s. Only electrons can cross the multiple trap energy levels and return to the valence band for carrier recombination. Thus, no free carriers can be measured, and the value at this time is the initial background value. In Figure 5b, the number of seconds of periodic illumination continues to increase because the carrier does not have sufficient time to return to the valence band for carrier recombination. When the light is turned on, the amount of carrier generation is greater than that of the carrier. The amount of subrecovery is turned on immediately after the next round has not been fully recovered. Free carriers that have not yet been recombined are added with the same amount of carrier generated by light. As the time in each round is increased, the curve shows increasing amplitude.

Figure 5c,d present the I-T and I-V diagrams of the bias value, rising current, and decay current value from +1.0 V to −1.0 V, respectively. In Figure 5c, when a positive bias voltage is applied to the component, the larger the bias value is, the greater the photogenerated current will be. When applying a negative bias voltage, the smaller the bias value is, the smaller the photogenerated current will be. This finding reflects the positive bias value. Figure 5d demonstrates the photocurrent value and bias voltage graph of the original element at each bias value for the 10th second of light-on and the 10th second of light-off in each period. As the bias voltage increases, the current rises and decays. The greater the current gap is, the greater the increase in the current value will be when compared to the decay current. In the negative bias voltage, as the bias voltage increases, the gap between the rising current and the decaying current becomes larger; however, the rising current value is smaller than the decay current value.

#### 3.2.1. P-Type Structure for Detection of Glutathione

The P-type structure transfers MoS_2_ onto P-Cu_2_O, in which Cu_2_O is not completely oxidized, and Cu_2_O remains between the structures. Thus, the cross-doping phenomenon of Cu_2_O and MoS_2_ occurred in the thin film surface. The E_C_ and E_V_ of cuprous oxide are about −1.91 and 0.29 eV, respectively, while those of copper oxide are about −1.01 and 0.41 eV, respectively [54]. The components are illuminated by electrons. Transition occurs from the valence band to the conductive band. After the electrons and holes are transmitted due to the energy level difference, the holes react with characteristic substances on the surface. To verify the detection mechanism of the P-type photoelectrochemical biosensor, we used GSH and GSSG as characteristic substances to measure the element. The experimental method is as follows. First, use sucrose solution and two characteristic substances of GSH and GSSG to prepare a detection solution with a molar concentration of 5 mM, and then use the light response system set up in this experiment to measure and observe the original components of this structure, and drop them into GSH. Determine the difference between the light switch and the GSSG characteristic solution. Figure 6a shows that the P-type photoelectrochemical biosensor has a larger response to GSH.

#### 3.2.2. N-Type Structure for Detection of Glutathione

The N-type structure is a heterojunction structure synthesized using N-ZnO nanorods and P-Cu_2_O crystals. The carrier type of the material in contact with the cell solution is N-type, which generates electrons and electricity after being illuminated. Electrons will accumulate during electron transport on the side of the N-type ZnO due to the difference in energy gap. If the GSSG characteristic solution that attracts electrons is dropped on the structure, the electrons will participate in the reduction reaction of GSSG. The higher the concentration of GSSG is, the stronger the ability to separate electrons from holes and the higher the light response will be. To verify the detection mechanism of the N-type photoelectrochemical biosensor, we used two characteristic substances, namely, GSH and GSSG, to measure the element. The measurement method is the same as the detection mechanism of the above P-type element [Figure 6b]. The response correlation of the N-type photoelectrochemical biosensor to GSSG is greater than that to GSH.

#### 3.2.3. P-Type Structure for Distinguishing Cancer Cell Stage

In this study, the MoS_2_/P-Cu_2_O structure was used to measure and analyze the number of esophageal cancer cells and modulated cells with different stages of cancer. We mainly measured four esophageal cancer cell lines: the first stage OE21-1 and the fourth stage OE21-4 of the Human Caucasian OE21 line as well as the first stage CE81T-1 and the fourth stage CE81T-4 of the Taiwanese CE81T line. After replacing the cell culture medium with sucrose, the photocurrent value of the four cancer cells was measured to adjust the number of cells: 1000, 5000, 10,000, and 15,000. As the measured GSSG content is greater, in principle, the greater the number of cells is, the lower the photoelectric response value should be. As shown in Figure 7a, the number of cells is likely to decrease corresponding to the photocurrent value. The error values of the four cancer cell lines at 10,000 and 15,000 are relatively similar.

In the experiment, the four cell lines were all fixed and used to extract about 5000 cells. The solution has a volume of 50 µL and a fixed bias value. In principle, for advanced cancer stages, the internal oxidative stress of cancer cells at the fourth stage would be more serious, leading to higher GSSG content. Cancer cells in the first stage have higher levels of GSH. We mentioned earlier that the sensitivity of P-type materials for measuring GSH is far greater than that for measuring GSSG, which means that P-type materials tend to measure GSH characteristic substances. The measurement results are presented in Figure 7b. The molybdenum component will enhance the photoelectrochemical signal due to the self-enhanced light absorption of molybdenum disulfide. Figure 7c,d show the measured photocurrent response values of esophageal cancer cells of the OE21 line and the CE81T line, respectively. Cells with a higher degree of cancer have a lower light response.

#### 3.2.4. N-Type Structure for Distinguishing Cancer Cell Stage

Four esophageal cancer cell lines were measured using the n-ZnO/p-Cu_2_O heterostructure: the zero stage OE21 and the first stage OE21-1 of the Human Caucasian OE21 line, the first stage of CE81T2-1, and the fourth stage CE81T2-4 of the Taiwanese CE81T line. As shown in Figure 8a–d, the photocurrent of OE21 was about 158% higher than that of the biosensor without that cell serum. For OE21-1, the photocurrent showed an outstanding value, that is, 223% higher without adding OE21-1. These results were also found in the two types of cancer, namely, CE81T2-1 and CE81T2-4, where the photocurrent increased compared with that in the absence of cancer cell sera; the results are 78% and 308% for CE81T2-1 and CE81T2-4 cancer cells, respectively. 

This result is consistent with the stages of ECC lines. The level of canceration of OE21-1 is greater than that of OE21, and the current value detected by the biosensor exhibits this trend. The photocurrent of CT81T2-4 outperformed that of CT81T2-1; CT81T2-4 was more severe in cancer level than CT81T2-1. This result validates the hypothesis of this study. When cancer becomes more severe, the concentration of GSSG in cancer cells is higher, and the photocurrent generated by the biosensor is larger.

## 4. Conclusions

The results may have implications for the use of N-type and P-type PEC biosensors to detect different stages of ECCs. The P-type biosensor on the device surface has a higher response to GSH. These results agree well with existing studies when the surface of the material has a P-type Cu_2_O structure; that is, the higher the concentration of the GSH characteristic solution is, the higher the photocurrent will be. The relationship between concentration and photocurrent indicates a linear growth trend [53,54]. The N-type biosensor has a higher response to GSSG. When cancer cells become more severe, the measured photocurrent will increase [37]. Based on the results, the more severe the cancer is, the higher the photocurrent measured by the P-type sensor will be. Given that normal cells are subjected to oxidative stress, the DNA structure is gradually damaged under the attack of reactive oxygen species (ROS), thereby changing the ratio of GSSG/GSH in normal cells. Thus, the invasion of ROS may cause more severe effects on the cells. When the concentration of GSSG increases, the photocurrent response detected decreases [55].

In future work, scholars should focus on particular aspects of tracking the primary tumor. A long-term follow-up of the study showed that the measured data will be archived into a large database, and a humanized display interface is designed through artificial intelligence. This tool can be used in the medical field to provide physicians with a general-oriented verification result. It may also provide doctors and patients with clear insights and help in intuitively understanding the diagnosis of cancer, particularly in esophageal cancer. Hopefully, it will be able to build a bridge for common communication between physicians and patients, and the vision of spreading cancer knowledge to the public and achieving prior detection and appropriate treatment will be just around the corner.

## Figures and Tables

**Figure 1 nanomaterials-11-01065-f001:**
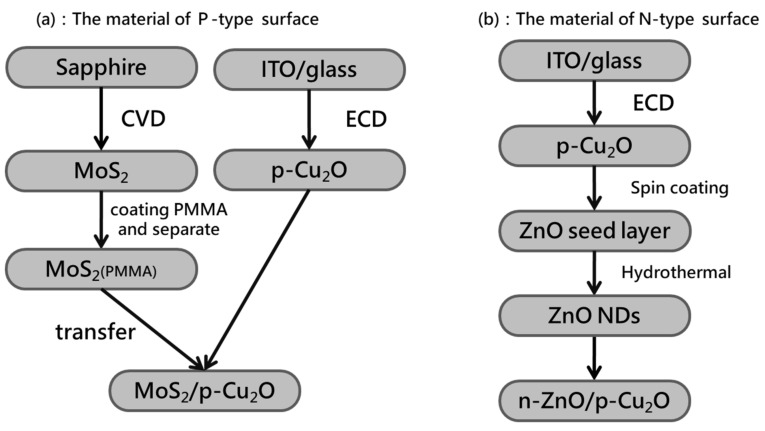
Preparation of photoelectrochemical biosensor components: (**a**) P-type and (**b**) N-type.

**Figure 2 nanomaterials-11-01065-f002:**
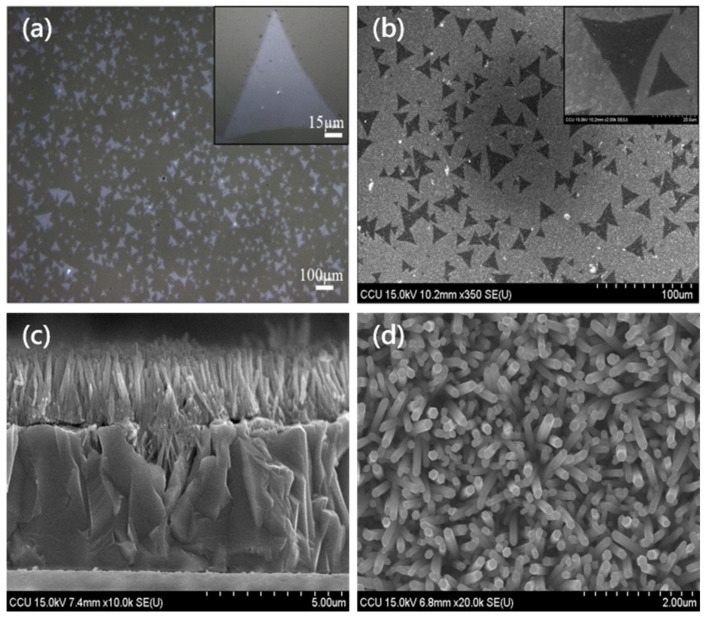
Component surface topography: (**a**) OM image of MoS_2_ grown on a sapphire substrate; (**b**) SEM top view of MoS_2_ transferred to Cu_2_O; (**c**) cross-sectional view of ZnO/Cu_2_O heterojunction; and (**d**) surface topography of ZnO nanorods.

**Figure 3 nanomaterials-11-01065-f003:**
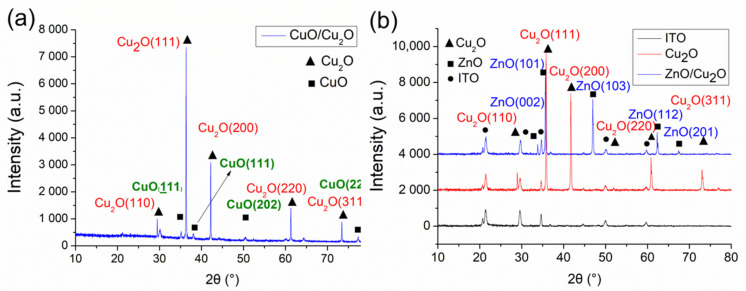
XRD patterns of (**a**) CuO/Cu_2_O and (**b**) fabricated ZnO/Cu_2_O heterostructure.

**Figure 4 nanomaterials-11-01065-f004:**
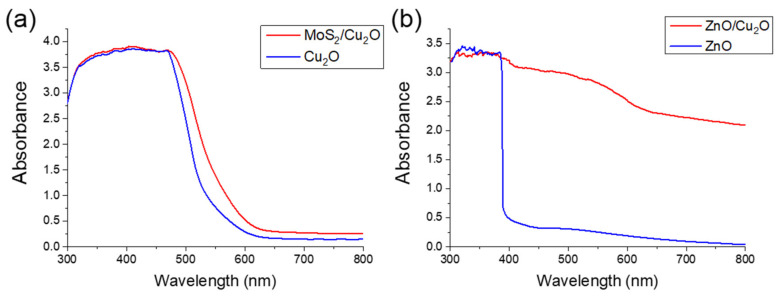
Ultraviolet–visible light spectrum analysis diagram of (**a**) MoS_2_/Cu_2_O and (**b**) ZnO/Cu_2_O.

**Figure 5 nanomaterials-11-01065-f005:**
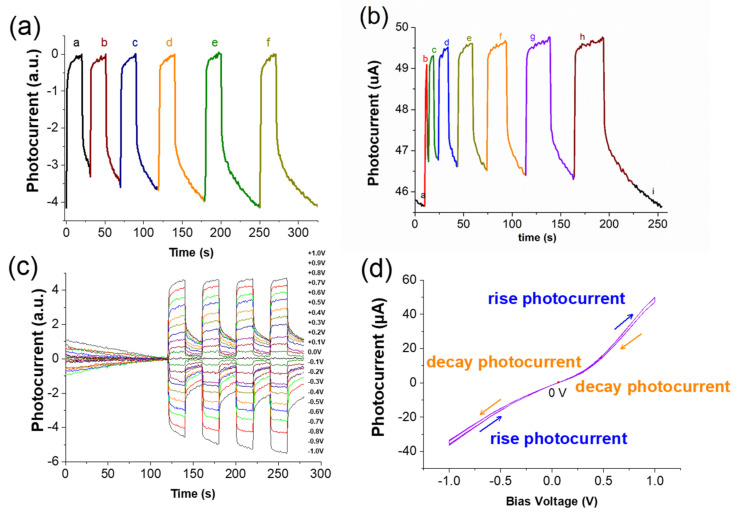
Component characteristic analysis: (**a**) the light time is 20 s, and the light is turned off for a. 10 s, b. 20 s, c. 30 s, d. 40 s, e. 50 s, f. 60 s. (**b**) a. photocurrent background value, b. turn on/off the light for 1 s, c. 5 s, d. 10 s, e. 15 s, f. 20 s, g. 25 s, h. 30 s, i. turn on/off 30; it takes another 30 s to decay the current to return to the original background value after seconds; (**c**) the original element photocurrent-time graph; and (**d**) the original element photocurrent-voltage graph.

**Figure 6 nanomaterials-11-01065-f006:**
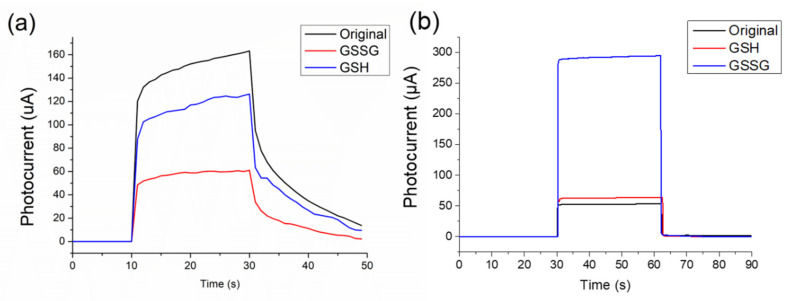
Sensitivity analysis of (**a**) light response of P-type element to GSH/GSSG and (**b**) light response of N-type element to GSH/GSSG.

**Figure 7 nanomaterials-11-01065-f007:**
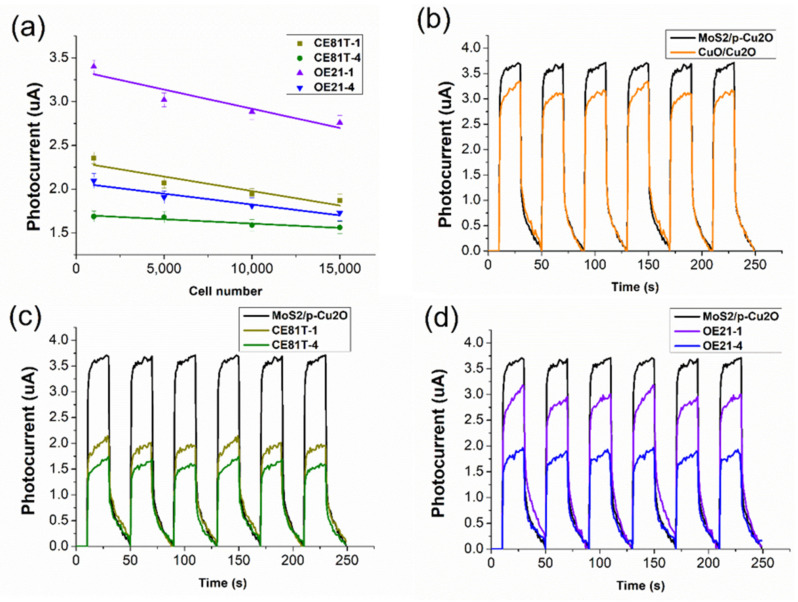
Photoelectrochemical response of esophageal squamous cell carcinoma: (**a**) P-type element modulates the linear relationship between the number of four cancer cell lines and light response analysis. (**b**) Comparison of light response analysis between MoS_2_/Cu_2_O and original elements; (**c**) MoS_2_/Cu_2_O and CE81T-1 and CE81T-4; and (**d**) MoS_2_/Cu_2_O, OE21-1, and OE21-4.

**Figure 8 nanomaterials-11-01065-f008:**
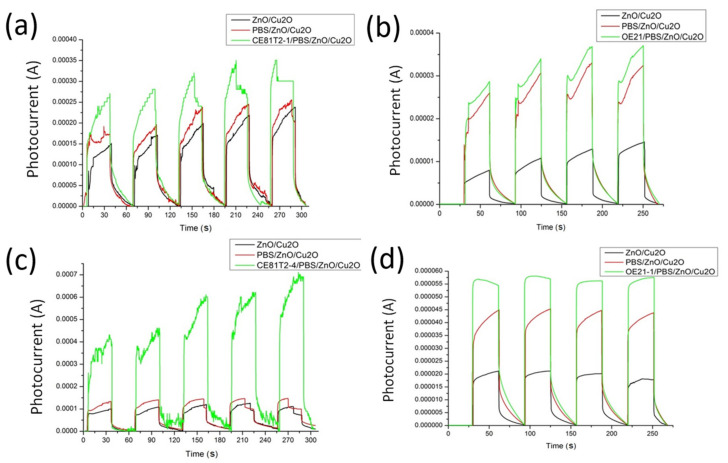
Photoelectrochemical response of esophageal squamous cell carcinoma by N-type element. (**a**) Comparison of light response analysis between blank biochip and adding PBS and PBS/CT81T2-1, respectively; (**b**) light response of empty biochip with PBS and PBS/OE21; (**c**) light response of empty biochip with PBS and PBS/CT81T2-4; (**d**) empty biochip with PBS and PBS/OE21- 1 light response.

## Data Availability

Data available on request due to restrictions eg privacy or ethical. The data presented in this study are available on request from the corresponding author. The data are not publicly available due to biocompatibility.

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
