# Peer review of "Characteristics of P-Type and N-Type Photoelectrochemical Biosensors: A Case Study for Esophageal Cancer Detection"

_nanomaterials, 2021, doi:10.3390/nano11051065_

Round 1

Reviewer 1 Report

In this manuscript of nanomaterials-1167005 “Characteristics of P-type and N-type Photoelectrochemical Biosensors: A Case Study for Esophageal Cancer Detection”, the author developed photoelectrochemical biosensors to detect esophageal cancer. Overall the topic is interesting and valuable but several points should clarify. Therefore, I recommend its publication in Nanomaterials after major revision.

What is GSSG means? The authors should explain the term before use an abbreviation.

Is there specific reason to select GSH and GSSG as representative materials of esophageal cancer?

Limit of detection ranges of both GSH and GSSG are important and what is the limit of detection ranges of both p-type and n-type sensors?

What is clinically meaningful range of GSH and GSSG?

The authors perform Sensitivity analysis of both light response of P-type element to GSH/GSSG and and light 217 response of N-type element to GSH/GSSG. Why the authors only used P-type biosensors for the analysis?

Figure 5 and 7 should be double column, not a single column. It is difficult to compare and understand the data. Additionally, there is no Figure 6 in the manuscript and two Figure 5. These should be revised.

Figure 5 (I assume figure 6) and 7 are somewhat confusing to understand. The authors partially investigate the component characteristic analysis for only used p-type. What about the n-type sensors? After Figure 6, suddenly n-type sensor is investigated for cancer cell stage. I could not find how 3.2.4 and 3.2.5 are connected and important. Therefore, the authors should be revised the overall manuscript and need to be clarified and explained.

Moreover, is there any evidence that the as-developed sensors can only specifically detect the GSH and GSSH, not for other molecules?

Overall, the authors claimed that n- and p-type biosensor to detect the cancer but there are no clear molar concentration range of target molecules. Additionally, the detail detection mechanism need to be explain to support the claim. In the biosensors, specificity and selectivity are major components that are needed to be verified.

Reviewer 2 Report

The authors describe an interesting nanostructured assemble for GSH/GSSG detection. It is a very concise and direct paper. Most of the concerns that I have, are regarding formal aspects, with no major scientific problems detected.

Lines 41-43
The authors say “GSH plays an important role in living organisms because it provides considerable biological information.” Although I understand why this is said, GSH plays an important role by, for example, helping on the redox homeostasis. Sure, GSH levels can give information (to us), as mentioned in lines 43-45 of the same page.

Lines 86-88
It would be great for the sake of uniformity and possible reproducibility, that amounts of reagents used can be indicated, as it was done, for example, for MoS2 growing (page 2, lines 92-94).

Line 100
A more suitable phrasing (more rigor, less redundance) for “sodium hydroxide with a molar concentration of 2 M.” could be “sodium hydroxide solution with a concentration of 2 M.”

Line 110
Please properly format the C6H12N4 chemical formula.

Figure 2 (a)
Please improve the readability of the values associated to the scale bars.

Lines 145-147
Please clarify what is considered dry air and humid air (ranges of air humidity %, for example).

Figure 3 (a)
To avoid confusion between peaks (for example between CuO(111) and CuO2(200) peaks), please consider rotate the red and green labels 90º counterclockwise. The same will also benefit figure 3 (b).

Figure 4
Absorbance does not have arbitrary units. It does not have units at all (see paper with the DOI 10.1021/jz4006916).

Figures 3, 4, 5, 7 (b-d)
I understand the purpose of make the lines visible, but I think the authors should come with a solution other than include negative values in the y axis.

Figures 5, 6 (a-b), 7 (a-d)
I understand the purpose of make the lines visible, but I think the authors should come with a solution other than include negative values in the x axis.

Figures 5, 6 (a-c), 7 (b-d)
Please change the unit in x axis from “sec” to “s”.

Figure 7 (e-h)
Please change the unit in x axis from “S” to “s”.

Figure 6
Figure 6 is wrongly labelled as “Figure 5” (line 261).

Figure 6 (d)
Please change the unit in x axis from “Volt” to “V”.

Figure 7 (a)
Please correct x axis label (“(Unit)”).

Figure 7 (e-h)
Please apply the same formatting and units of b-d charts from the same figure.

Reviewer 3 Report

This is an interesting study and important for the cancer detection field that uses sensing tools.

Abstract can be modified and rewritten. 'Certain level of discrimination' and 'small amount of glutathione' sentences can be revised and could be more specific.

Any confocal imaging based data to support the localized interaction between the biorecognition element to cancer cells? 

The data collected and presented in the form of graphs in results section is quiet comprehensive and supports the claims made. Overall this is a good paper and can be accepted with minimal revision.

Reviewer 4 Report

The authors have done a good job characterizing a pair of photosensors for use in detecting GSH and GSSG concentrations, but a more thorough discussion of these compounds is needed, and the methods sections needs to be made more clear and robust.  My specific comments below address where I think the paper could be improved:

  • A better link between GSH and esophageal squamous cells should be established in the introduction.  What is the expected concentration of GSH supposed to be in relation to healthy cells?  How would it be employed in active detection?  Is GSH expected to be elevated or depressed in esophageal cancer patients?
  • Materials and Methods needs a section to go over where the chemicals and supplies were acquired from.
  • Subsections of Materials and Methods are misnumbered.
  • Cell Culture section of methods needs concentrations specified for all solutions.
  • Section 3.1.2 – Do the peak areas at each angle correspond to those expected for the materials?
  • The Cu2O layer greatly increases absorbance in the visible range when bonded to ZnO, but far less so when mixed with MoS2, is there a reason for this?
  • Section 3.2.1 – Part of this needs to be moved into materials and methods
  • The first paragraph of section 3.2.3 would help in understanding the tests performed in section 3.2.1, and should be moved ahead of section 3.2.1
  • Parts of section 3.2.4 and 3.2.5 should be moved to the methods section.
  • How do you know that the components being measured are GSH and GSSG and not other components of the cell line.
  • How can this type of test be used to screen cancer in people? Is the concentration of GSH or GSSG in patient serum expected to change in cancer patients versus healthy patients?

Round 2

Reviewer 1 Report

The manuscript is now ready to publish in this journal

Author Response

Thank you for your comments.

Reviewer 4 Report

Thank you for your response, and addrerssing some of my concerns.,  I however think that several areas still need to be fixed before I can recommend publication:

Hello,

  1. You still have not described the expectant concentration of GSH/GSSG in the introduction. You have only stated it is linked to cancers, but have not described the expected concentration versus healthy individuals.  I see you have added some of this info to the conclusions but it should really be in the introduction.
  2. This is not what I meant, I mean a section of where materials were purchased from. For example here is such a section:

2.1 Materials and Components

Unless otherwise specified, all reagents including LPA and b-casein were purchased from Sigma Aldrich, Oakville, Ontario. Gelsolin protein was produced by expression from PSY5 plasmids containing the gelsolin gene with a histidine tag as described previously [27]. The plasmids were kindly provided by Professor Robert Robinson of the University of Singapore. Protein mass and purity were determined by SDS-PAGE (12% acrylamide) and concentration by absorbance at 280 nm. Actin from rabbit muscle was purchased from Alfa Aeser or Sigma Aldrich and modified as outlined previously [27]. Protein concentration was determined by absorbance at 280 nM. Quartz crystals (AT-cut, 13.5 mm in diameter, 20 MHz fundamental frequency) were purchased from Lap-Tech Inc., Bowmanville, Ontario.

  1. Thank you for making this change.
  2. Add this response to the paper itself, just telling me doesn’t help other readers to understand this.
  3. Section 3.1 is not the materials and methods, parts of section 3.2.1 need to be moves into section 2 as much of it describes methods that were used. Additionally there are many techniques used in the results such as SEM and X-ray diffraction that were not mentioned in the methods section.  Methods for these techniques need to be presented in section 2; what microscopes were used, what settings were used, etc…
  4. Thank you for making this change.
  5. Again, you describe methods used in these results sections that were not mentioned in the methods section. All methods used in the paper need to be described in the methods section, not in the results.
  6. This sentence: “We acknowledge that there are many redox-active molecules in a cell, so further research is needed to understand the contribution of different redox-active molecules” should be included in the paper as it is still unclear how the values you measured are known to be a result of GSG and not something else.
  7. Thank you for adding this information to the paper. I think it would be better suited to the introduction of the paper as it establishes how such screening would work, and gives credence to the development of this sensor.

Round 3

Reviewer 4 Report

Thank you again for making many of the suggested changes to your paper.

I think you misunderstood when I suggested parts of the results describing the microscope work and x-ray diffraction work needed to be moved to the methods section as they describe methods.  You moved the entire section, when just the methods of the section should have been moved (the highlighted parts in what is now sections 2.3.1, 2.3.2, and the sentences in 2.3.3 describing the instrument setup should be in the methods section.  The remainder of the section, and the presented figures should have been kept in the results section as they are results to the paper.

If you could make this formatting change so that the methods and results of the paper are clear and correctly placed in their respective sections I would be happy to recommend publication.

Author Response

We created the new section “Optical and material analysis instruments” as 2.4. Then we moved the highlighted parts of sections 2.3.1, 2.3.2, and 2.3.3 to new section 2.4. We restore the remaining chapters to their original positions. In the layout of the article, we put the figure 2 in front of the section 3 for a more compact layout.